# An Overview of Oil-Mineral-Aggregate Formation, Settling, and Transport Processes in Marine Oil Spill Models

**Xiaomei Zhong [1], Haibo Niu [2,\*], Pu Li [3,4,\*], Yongsheng Wu [5] and Lei Liu [1]**

1. Department of Civil and Resource Engineering, Faculty of Engineering, Dalhousie University, Halifax, NS B3H 4R2, Canada; xm976877@dal.ca (X.Z.); lei.liu@dal.ca (L.L.)
2. Department of Engineering, Faculty of Agriculture, Dalhousie University, Truro, NS B2N 5E3, Canada
3. School of Marine Science, Sun Yat-Sen University (Zhuhai Campus), Zhuhai 519082, China
4. Pearl River Estuary Marine Ecosystem Research Station, Ministry of Education, Zhuhai 519082, China
5. Ocean and Ecosystem Science Division, Fisheries and Ocean Canada, Bedford Institute of Oceanography, Dartmouth, NS B2Y 4A2, Canada; yongsheng.wu@dfo-mpo.gc.ca
\* Correspondence: haibo.niu@dal.ca (H.N.); lipu8@mail.sysu.edu.cn (P.L.)

**Abstract:** An oil spill is considered one of the most serious polluting disasters for a marine environment. When oil is spilled into a marine environment, it is dispersed into the water column as oil droplets which often interact with suspended particles to form oil-mineral-aggregate (OMA). Knowing how OMA form, settle, and are transported is critical to oil spill modelling which can determine the fate and mass balance of the spilled volumes. This review introduces oil weathering and movement, and the commonly used numerical models that oil spill specialists use to determine how a spill will evolve. We conduct in-depth reviews of the environmental factors that influence how OMA form and their settling velocity, and we review how OMA formation and transport are modelled. We point out the existing gaps in current knowledge and the challenges of studying OMA. Such challenges include having to systematically conduct laboratory experiments to investigate how the environment affects OMA formation and settling velocities, and the need for a comprehensive algorithm that can estimate an OMA settling velocity.

**Keywords:** oil-mineral-aggregate (OMA); oil spill modelling; OMA settling velocity; OMA transport modelling

## 1. Introduction

Although it is critically important to reduce the consumption of petroleum hydrocarbons, oil products are still important to society due to the lack of alternative energy sources such as fuel for marine transportation. Producing and transporting oil products means that oil spills can potentially occur due to accidents and/or operational errors [1]. Spills of crude oils and other petroleum products in a marine environment is a major environmental problem. A report by the International Tanker Owners Pollution Federation (ITOPF) stated that there were about 10,000 large oil spills (>7 tonnes) from tankers and oil carriers from 1970 to 2018 [2]. Although the frequency of large spills has decreased over the past five decades [2], the number of small spills (for example, the *Marathassa* oil spill in English Bay, British Columbia in 2008) is still high [3]. These small oil spills usually occur in the open ocean and are caused by inappropriate onboard operations. Furthermore, because of their relatively small environmental impact, the small spills receive less public concern than large oil spills [4]. However, regardless of their size, oil spills are some of the most critical forms of marine pollution, leading to significant long-term impacts on the environment and socio-economy of affected areas. Therefore, local authorities and oil producers develop contingency plans to prevent, control, and reduce the negative impacts of oil spills. Accurately predicting oil spill behavior is critical for an immediate, quick, and efficient spill response.

Oil spill modelling involves simulating oil spill trajectories and behavior based on mathematical algorithms controlled by hydrodynamics and wind forcing. Once oil is spilled into water, two common processes occur simultaneously: weathering, which alters the physico-chemical properties of the spilled oil; and transport, which increases the size of the contamination area. Three factors affect these two processes: the volume of oil spilled, where the oil is spilled, and the type of oil [5]. Numerical modelling is commonly employed to predict the fate and behavior of spilled oil, and modelling is an important aspect of oil spill contingency planning and coastal management. For example, French-McCay et al. [6] used the Spill Impact Model Application Package (SIMAP) to simulate the trajectories and fate of the Deepwater Horizon oil spill. Over the past few decades, other numerical models have been developed to simulate oil spills in marine environments including: OSCAR (Oil Spill Contingency and Response) model by SINTEF (*Stiftelsen for INdustriell of TEknisk Forskning ved NTH*—Foundation for Industrial and Technical Research) [7,8], OILMAP and SIMAP by RP-ASA (Applied Science Associates, Inc., Narragansett, RI, USA) [9,10], the MOHID water model by MARETEC (Marine and Environmental Technology Research Center) [11], GNOME (General NOAA Operational Modeling Environment) by NOAA (National Oceanic and Atmospheric Administration) [12], and OpenDrift model and its subclass, OpenOil, by the Norwegian Meteorological Institute [13,14].

When oil is spilled into a marine environment, a significant portion of the oil is dispersed into the water column as oil droplets [15,16]. These oil droplets interact with suspended sediments to form oil-mineral-aggregate (OMA) as shown in Figure 1. OMA have been observed to form in many oil spills. For example, in 2010, more than 3.2 million liters of diluted bitumen spilled from the Enbridge Line 6B pipeline into the Kalamazoo River in Michigan, and a large amount of spilled oil (heavy component) interacted with the sediments to form OMA along channel margins, backwaters, and oxbows [17,18]. The formation, settling, and transport of the OMA are key to the transport and fate of the spilled oil [19–21].

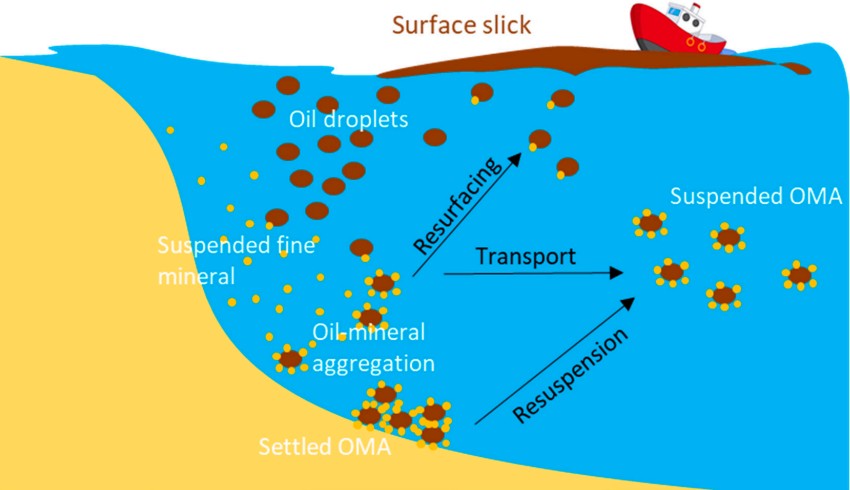

**Figure 1.** How OMA form in the nearshore environment (reproduced from Gustitus et al. [22], with kind permission Copyright 2020 from *John Wiley & Sons, Inc.*).

Although several reviews on oil spill modelling and studies about OMA have been published in recent years, an updated and comprehensive review for OMA formation, settling, and transport is lacking. Our paper, therefore, aims to provide a dedicated review of oil spill trajectory modelling, OMA formation and settling processes, and modelling of OMA processes. This review is organized as follows: (1) Section 2 describes oil weathering and movement, and commonly used oil spill models to give an overview of current oil spill modelling; (2) Section 3 details how OMA form and settle, and we describe how OMA formation and transport have been and are currently modelled; (3) Section 4 discusses existing challenges and future work on modelling OMA formation, settling, and transport;

(4) in Section 5, we summarize our review with key findings. Figure 2 illustrates the structure and themes of this review.

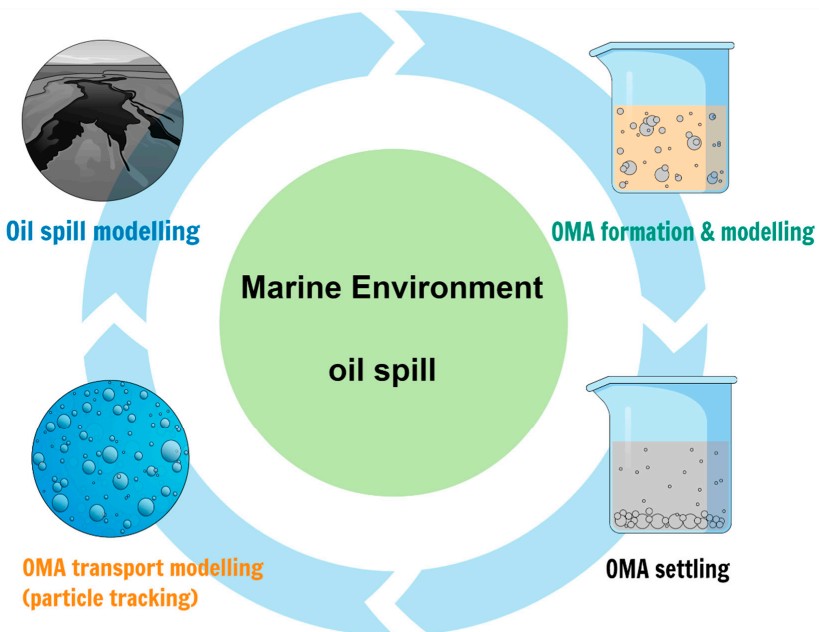

**Figure 2.** General schematic of this review.

## 2. Oil Spill Weathering and Movement and How They Are Modelled

### 2.1. Oil Weathering

Oil weathering refers to oil spreading, evaporation, emulsification, and natural dispersion, followed by dissolution, photo-oxidation, biodegradation, and sedimentation [23]. Each process happens at different times at varying rates. For example, oil evaporation begins immediately when the oil slick first presents on the water surface. Oil evaporation is a relatively short process. In contrast, oil emulsification is a long process that begins relatively slowly [24]. Furthermore, different weathering processes have different impacts on the physico-chemical properties of the spilled oil. Among these processes, photo-oxidation and biodegradation are the most critical for modifying the chemical compositions of spilled oil [23,25]. Furthermore, as evaporation decreases the volume of the spilled oil, it also increases the oil's density [26]. After evaporation, the remaining heavy components of a lighter oil can combine with sediments in the water column and subsequently sink to the bottom. The oil remaining at the surface naturally disperses into the water column [27]. Lastly, oil emulsification depends on the types of oil and conditions in the surrounding environment. Emulsification changes the oil's physical properties such as water content and viscosity [28].

### 2.2. The Movement of Spilled Oil

Knowing how spilled oil moves on the water surface is important to oil spill modelling. Spreading is one of the most critical movement processes because spreading oil can increase the size of the contaminated area [24]. Furthermore, gravity and oil-water interfacial tension can help the spilled oil spread into a slick over the water surface [23]. The effects of gravity could gradually diminish over time, but the oil would continue to spread under the effects of interfacial tension [29]. The spreading rate of spilled oil is closely related to environmental conditions, such as tidal streams, water currents, coastlines, and wind speeds, as well as being related to oil properties [24,30–32]. For instance, in a confined waterway, it is difficult for spilled oil to spread quickly to reach the shoreline [33], but, in the open ocean, spilled oil can spread rapidly and widely to cover a large area in a short period [24]. Moreover, spilled oil that has lower molecular weight components presumably spreads more quickly [24]. Additionally, spreading depends on oil weathering.

For example, evaporation and dissolution can reduce the area covered by the spreading oil [32].

Oil advection at the sea surface and into the water column is another crucial movement process and mainly depends on ocean currents, wind, and waves. The effects of the wind and currents on spilled oil depend on the spill location and hydrographic conditions.

### 2.3. Commonly Used Models

Various oil spill models have been developed to track the fate and behavior of spilled oil. These oil spill models are classified as oil weathering models or trajectory/deterministic models. Oil weathering models (OWM) do not consider oil transport and only simulate oil weathering processes.

The ADIOS2 (Automated Data Inquiry for Oil Spills, Silver Spring, MD 20910, USA) oil weathering model, which developed by NOAA, mainly includes the weathering processes of evaporation, dispersion, emulsification, spreading, and sedimentation. ADIOS can also assess how effective clean-up techniques would be [34,35].

The IKU Oil Weathering Model, which developed by SINTEF, is also commonly used. The IKU OWM is also known as SINTEF OWM. SINTEF OWM is based on small and mesoscale laboratory tests and full-scale field experiments [7,8], and SINTEF OWM considers emulsification, natural dispersion, and evaporation as its main weathering processes. The model applies a pseudo-component method to calculate how much oil is lost to evaporation, and Mackay's equation evaluates the viscosity of the emulsified oil [36,37]. It is also worth mentioning that nearly 200 different types of oil have been characterized by the SINTEF laboratory in Norway, all of which are included in SINTEF OWM [38].

More specialized weathering models, such as the Diluted Bitumen Weathering Model (DBWM), have been developed and recently validated against meso-scale experimental data to simulate the weathering of unconventional crude products such as diluted bitumen [39].

Trajectory/deterministic models are grouped as two-dimensional (2D) and three-dimensional (3D) models [40]. A 2D model only simulates the trajectory of spilled oil at the water's surface, whereas a 3D model tracks the spilled oil at the surface and in the water column. Various algorithms have been developed for trajectory/deterministic models, including Eulerian methods, Lagrangian methods, and Eulerian–Lagrangian methods (ELMs). According to state-to-the-art reviews [40–42], Lagrangian methods are the most popular algorithms for 3D oil spill models. For instance, the COSMoS model developed by Environment and Climate Change Canada [43] and SPILLCALC developed by Tetra Tech EBA [44] use Lagrangian algorithms. Table 1 lists examples of 3D models that use Lagrangian algorithms.

**Table 1.** Examples of 3D oil spill models that use a Lagrangian algorithm to simulate oil transport.

| Model | Developer | References |
|---|---|---|
| COSMoS | Environment and Climate Change Canada (ECCC) | [43] |
| COZOIL | Department of the Interior Minerals Management Service | [45] |
| GNOME Suite | National Academies of Sciences, Engineering, and Medicine (NOAA) | [46] |
| GULFSPILL | KFUPM/RI | [47] |
| MEDSLIK/MEDSLIK-II | Oceanography Centre of the University of Cyprus (OC-UCY) | [48–50] |
| MIKE 21/3 | Danish Hydraulic Institute (DHI) | [51] |
| MOHID | MARETEC (Marine and Environmental Technology Research Center) | [52] |
| MOTHY | Météo-France | [53] |
| OILMAP/SIMAP | ASA | [54,55] |
| OILTRANS | The Atlantic Regions' Coastal Pollution Response (ARCOPOL) | [56] |
| OpenDrift/OpenOil | Norwegian Meteorological Institute | [13,14] |
| OSCAR | SINTEF | [57] |
| OSRA | Bureau of Ocean Energy Management (BOEM) | [58–60] |
| POSEIDON-OSM | Hellenic Centre for Marine Research (HCMR) | [61,62] |
| SPILLCALC | Tetra Tech EBA | [44] |

Oil spill models can also be classified based on their purpose. After the Deepwater Horizon oil spill (2010 to 2020), Ainsworth et al. [63] categorized and summarized 330 published modelling applications developed by the Gulf of Mexico Research Initiative (GOMRI), the Natural Resource Damage Assessment (NRDA), and other researchers. Ainsworth et al. classified the models by (1) circulation/mixing, (2) abiotic transport (far field), (3) oil fate, (4) biotic transport, (5) biological impacts, (6) other plume dynamics, (7) turbulence/local mixing, (8) water chemistry, (9) atmosphere, (10) oil spill response support, and (11) other [63]. Furthermore, Keramea et al. [64] summarized commonly used models based on model capabilities and purpose. Keramea et al. classified the oil spill models into: (1) surface oil spill models and blowout/buoyant plume models, (2) operational response models, (3) deep sea blowout/buoyant plume models, and (4) models with spill response and environmental impact assessment support [64]. Zhao et al. [65] evaluated algorithms for oil spreading, evaporation, emulsification, dispersion, dissolution, biodegradation, and photo-oxidation [65]. We note that sedimentation (OMA formation and transport) is absent in all the 2D and most 3D oil spill models. The 3D models that include sedimentation usually describe it with simple coefficients and do not have detailed sedimentation algorithms.

## 3. Oil-Mineral-Aggregate (OMA)

### 3.1. Factors Influencing OMA Formation

OMA formation is affected by many factors, including oil properties (type, droplet size, and concentration), particle properties (size and shape, organic matter content, density, and concentration), and ambient conditions such as mixing energy, salinity, and temperature [66]. It has been reported that the average aggregate size is positively correlated to the oil-particle ratio (g oil/g sediment) [67]. However, when only considering sediment concentration, the relationship between particle concentration and the average aggregate size fits a Gaussian distribution with the highest average aggregate size of ~1400 um at a particle concentration of 0.8 g/L [67]. Kehlifa et al. [68] showed that normalized cumulative size distributions of OMA are like the size distributions of mineral-free oil droplets formed in seawater.

In terms of the oil viscosity's influence on how OMA form, Wood et al. [69]) observed that the aggregation rate of OMA is greater as the oil viscosity and density decrease. However, Khelifa et al. [70] indicated that oil viscosity has a negligible effect on the maximum and mean sizes of OMA.

The types of clay in the suspended sediment impacts how OMA form. Zhang et al. [71] and Wang et al. [72] found that the size of an OMA increases as the minerals change from hydrophilic to hydrophobic. The OMA's size increases from a few micrometers for a natural kaolin (hydrophilic) to tens of micrometers for a modified kaolin (hydrophobic). The change in size occurs because the hydrophobicity of minerals promotes the affinity of minerals to oil, which enhances the formation of OMA. Khelifa et al. [68] also indicated that the clay type is crucial to the OMA's size, which has even more influence on aggregate size than the type of oil does.

The effect of salinity on how OMA form is controversial. Several studies illustrate that low salinity (0–5 ppt) strongly influences OMA formation, and no significant effect exists for greater salinity increases [68,73–75]. For instance, Le-Floch et al. [74] reported that the proportion of oil in an OMA increases as the salinity increases, but the oil content stabilizes when salinity is greater than ~2 ppt. Khelifa et al. [68] illustrated that the median and maximum size of OMA increases significantly as the salinity increases from 0 ppt to 1.2 ppt, but a further increase in salinity to 3.5 ppt decreases the OMA's size [68]. However, Payne [76] found the opposite result: an OMA forms more quickly with salinities of 15 and 30 ppt than in freshwater. Furthermore, Guyomarch et al. [77] reported that the OMA formation rate slows as salinity increases from 10 to 50 ppt. Theoretically, the effects of salinity on OMA formation occur because of the change in the electrical double layer thickness of the oil and mineral particles. As salinity increases, the layer thickness compresses, which

reduces surface repulsion and increases the inter-particle attraction between the oil and mineral. However, the influence of salinity on OMA formation is minimal once the layer thickness is below a threshold as the level of salinity in the water rises. Therefore, OMA formation eventually peaks and stabilizes as the salinity increases [74].

Mixing energy is an important environmental factor that influences the size of OMA [58,71,78–80]. An adequate mixing energy promotes the interaction between dispersed oil droplets and suspended particles so that OMA can form [79]. On the other hand, an OMA could break apart if the mixing energy is too aggressive [71]. Ji et al. [58] conducted a series of laboratory experiments to investigate the impact of mixing energy on the time it takes for OMA to form. They suggested that the collision efficiency between oil and sediments under a higher mixing energy rate (200 rpm) is three times higher than under a lower mixing energy rate (150 rpm). High mixing energy results in rapidly forming OMA during the first 3 h, followed by the OMA breaking apart. Ji et al. found that, after 24 h of mixing, only in low energy mixing cases would the dispersed OMA reform because there would be more oil droplets on the water's surface.

As a human initiated influence, chemical dispersants are key to OMA formation. Page et al. [15] carried out a series of wave tank experiments to study the effectiveness of using dispersants as an oil spill chemical countermeasure in the surf zone. Their experiment results indicated that chemically dispersed oil associates less intensively with mineral matter than physically dispersed oil [15]. Zhang et al. [71] found that applying chemical dispersants is a dominant factor in how OMA form and behave. Several wave tank experiments conducted by Lee et al. [81] indicated that chemical dispersants reduce surface tension between the oil and water and stimulate interaction between oil and fine mineral, eventually increasing the concentration of OMA. Lee et al. also indicated that the chemical dispersants reduce the size of OMA to a mean diameter of 15–25 μm [81]. However, Khelifa et al. [82] observed that chemical dispersants enhance the stickiness of oil to lead to larger OMA, probably due to the surfactant coating the oil droplets. Fu et al. [83] carried out roller table experiments to explore how marine oil snow forms when a stereotype oil dispersant (Corexit EC9500A) is present. Fu et al. found that adding the chemical dispersant enhances particle aggregation and formation of marine snow. O'Laughlin et al. [84] claimed that dispersants delay flocculation of natural sediments and create a surplus of available sediment to interact with oil droplets and potentially form OMA.

### 3.2. OMA Settling

The settling velocity of OMA mainly depends on their composition densities. The buoyancy of OMA should cause OMA to rise to the surface if the composition is mainly oil; otherwise, the OMA composed primarily of sediment should sink to the benthic layer [85]. A better understanding of OMA settling is critical to predicting the fate and transport of OMA.

Table 2 summarizes laboratory experiments of OMA settling. Muschenheim and Lee [86] investigated the amount and rates of OMA formation and settling velocities by using a focused flow reactor. They reported that large (100 to 200 μm) OMA have settling velocities ranging from 2.2 to 10.4 mm/s. Waterman and Garcia [87] tested settling velocities by using a 1.6 m height Plexiglass settling column and observed that the OMA had settling velocities between 1.0 and 11.2 mm/s, with the majority being between 1.0 and 3.0 mm/s. O'Laughlin et al. [84] conducted experiments to measure the variability in particle size and settling velocity of OMA in response to sediment concentrations and presence/absence of chemical dispersants. O'Laughlin et al. found that OMA size, settling velocity, and effective particle density increase under a higher concentration of suspended sediment (comparing 10 mg/L with 50 mg/L), which indicates that high concentration suspension produces large, inorganic-derived flocs that settle rapidly. Their results also suggested that dispersants inhibit natural sediment flocculation. Ye et al. [88] conducted laboratory experiments to investigate the influence of clay types on OMA structures and settling velocities by using the LabSFLOC-2 system and digital microscopy. Ye et al. found that

for low stickiness Kaolinite clay, droplet OMA form with much smaller settling velocities than the settling velocities of pure Kaolinite flocs. Furthermore, Ye et al. found that high stickiness Bentonite clay, generates flack/solid OMA with settling velocities greater (by 25%) than settling velocities of pure Bentonite flocs.

**Table 2.** Summary of laboratory experiments on the OMA settling velocity.

| References | Main Objectives/Methods | Results |
|---|---|---|
| [21] | To measure OMA settling velocities using a focused flow reactor | Settling velocities ranging from 2.2 to 10.4 mm/s for 100–200 μm OMA |
| [87] | Settling velocity tests conducted in a 1.6 m height Plexiglass settling column. | Settling velocities between 1.0–11.2 mm/s, with the most in the between 1.0 and 3.0 mm/s. |
| [84] | Sediment concentration influences settling velocity | Higher concentration of suspended sediment (10 vs. 50 mg/L) means greater settling velocity and effective particle density |
| [88] | To explore effects of clay type on OMA structure and settling velocity using the LabSFLOC-2 system and digital microscopy | For low stickiness Kaolinite clay, OMA settling velocity was about twice as small as for pure Kaolinite flocs; for high stickiness Bentonite clay, the OMA settling velocity was greater than for pure Bentonite flocs |

### 3.3. Modelling OMA Size Distribution

The size of OMA varies from a few micrometers to hundreds, even thousands, of micrometers [67,71,78,79]. The range of sizes of OMA illustrates the complexity of OMA formation. This complexity encouraged scientists to develop sophisticated numerical models to predict how OMA form and subsequently transported. However, modelling OMA formation and transport is still at an early stage and the ability of the models to accurately predict either process is limited. In most oil spill models, only the ratio of mineral-stabilized oil is computed from standard equilibrium partitioning theory (by using a dimensionless coefficient), and the concentration of suspended particle matter in the water column is just estimated [42]. Therefore, there is still a need to develop even more advanced OMA models that can be incorporated into existing oil spill models.

With regards to modelling OMA formation, the population balance equation, based on collision theory between oil droplets and suspended sediment materials, is currently widely used [68,89–93]. The population balance equation for particle collision due to Brownian motion was first proposed by Smoluchowski [94] and is listed as Equation (1) in Table 3. After Smoluchowski developed their equation, other physical mechanisms, such as shear turbulence and differential settling, were included in the modelling (Equation (2) in Table 3) [95]. Sterling et al. [92] extended the population balance equation to simulate the changes in particle size distribution and density due to aggregation (Equations (3) and (4) in Table 3). Sterling et al. conducted batch flocculation experiments and introduced an algorithm to estimate homogeneous collision efficiency values ($\alpha_{HOMO}$) to account for the effects of particle types on OMA formation. Furthermore, Sterling et al. found that $\alpha_{HOMO}$ is greater for clay (0.7) and crude oil (0.3) compared to silica (0.01); thus, they classified clay and crude oil as cohesive particles and classified silica as a non-cohesive particle. In addition, they found that $\alpha_{HOMO}$ is similar for oil-clay (0.4) and oil-silica (0.3), suggesting that crude oil increases the aggregation of non-cohesive silica particles [92]. However, the Sterling et al. model utilized a looped term to represent oil–oil, oil–particle, and particle–particle interactions, mainly because the laboratory operations probably could not distinguish between the interactions. Moreover, Sterling et al. did not include an OMA breakup process in their model.

**Table 3.** A summary of equations relevant to modelling OMA formation.

| Equation | Equations | Denote | References |
|---|---|---|---|
| (1) | $\theta = \frac{dn_k}{dt} = \frac{1}{2}\sum\limits_{i+j=k}\alpha\beta(i,j)n_in_j - n_k\sum\limits_{i=1}^{\infty}\alpha\beta(i,k)n_i$ | $\alpha$ is the collision efficiency. $\beta$ is the collision frequency, and $n_i$ and $n_j$ are the particle concentrations for the particles of size $i$ and $j$, respectively. | [94] |
| (2) | $\beta = \beta_{Br} + \beta_{sh} + \beta_{ds}$ $\beta_{Br} = \frac{2kT}{3\mu}\left(\frac{1}{d_i} + \frac{1}{d_j}\right)\left(d_i + d_j\right)$ $\beta_{sh} = \frac{G}{6}\left(d_i + d_j\right)^3$ $G = \left(\frac{\varepsilon}{\mu}\right)^{0.5}$ $\beta_{ds} = \frac{\pi}{4}\left(d_i + d_j\right)^2\left|\left(U_i - U_j\right)\right|$ | $\beta_{Br}$, $\beta_{sh}$, and $\beta_{ds}$ are the collision efficiency due to Brownian motion, fluid shear, differential sedimentation, respectively. $k$ is the Boltzman's constant. $T$ is the absolute temperature, and $\mu$ is the dynamic viscosity of the media. $d_i$ and $d_j$ are the effective particle diameters. $G$ is the root mean square of the velocity gradient. $\varepsilon$ is the dissipation rate. $U_i$ and $U_j$ are the settling velocities of the two collided entities of size $d_i$ and $d_j$. | [92,93,96] |
| (3) | $\frac{dn_{k,q}}{dt} = D_z\frac{\partial^2 n_{k,q}}{\partial Z^2} - w_{k,q}\frac{\partial n_{k,q}}{\partial Z} + \theta_{k,q}$ | $n_{k,q}$ is the particle size distribution. $k$ is particle volume. $q$ is effective density. $z$ is the vertical distance. $D_z$ is the vertical dispersion coefficient. $w_{k,q}$ is the settling velocity. $\theta_{k,q}$ is the interaction term due to coagulation. | [96] |
| (4) | $\alpha_{obs}(o,p) = \alpha_{HOMO,1}(sf_{1,o})(sf_{1,p}) - \alpha_{HOMO,2}(sf_{2,o})(sf_{2,p}) - \alpha_{HET,1-2}(sf_{1,o}sf_{2,p} - sf_{1,p}sf_{2,o})$ | $sf_{1,o}$, $sf_{2,o}$, $sf_{1,p}$, and $sf_{2,p}$ are surface fractions of constituent particles 1 and 2 in aggregate with density $o$ and constituent particles 1 and 2 in aggregate with density $p$. $\alpha_{HOMO,1}$, $\alpha_{HOMO,2}$, and $\alpha_{HET}$ are probabilities of successful aggregation through contacting floc constituent types 1-1, 2-2, and 1-2. | [92] |
| (5) | $\frac{\partial C_i}{\partial t} + \frac{\partial(u_kC_i)}{\partial x_k} = \frac{\partial\left(E_k\frac{\partial C_i}{\partial x_k}\right)}{\partial x_k} + w_{si}\frac{\partial C_i}{\partial x_3} + S_{i,Agg} + S_{i,De} + S_{i,Abs}$ | For i = 1–5, corresponding to 5 species mentioned in assumptions. k = 1, 2, 3 correspond to directions of $x$, $y$, $z$. $C_i$ = volumetric concentration of the $i$th species. $u_k$ = component of current velocity in $x$, $y$, and $z$ directions. $w_{si}$ is the buoyant/settling velocity of the $i$th species. $E_k$ is the diffusion coefficient in $k$th direction. $S_{i,Agg} = \frac{\partial C_i}{\partial t}\big|_{Agg}$ = source/sink terms of the $i$th species due to aggregation. $S_{i,De} = \frac{\partial C_i}{\partial t}\big|_{De}$ = sink term due to deposition of the $i$th species, and $S_{i,Abs}$ = source/sink term for the $i$th species due to partitioning. | [89] |
| (6) | $\alpha(t) = \alpha_{sta}\left(1 - \frac{\sum A_{p-proj}\ in\ OMAs}{F_{SP}\sum A_o}\right)$ | $\alpha_{sta}$ is the stability ratio. $A_o$ is the surface area of an oil droplet. $A_{p-proj}$ is the projection area of particles on the droplet surface when an OMA forms, and $F_{SP}$ is a factor to account for particle shape and packing effects on the coagulation process. | [93] |
| (7) | $P_{brk} = \begin{cases} 0, & n_b = 0 \\ 0.5, & n_b = 1 \\ 1, & n_b > 1 \end{cases}$ | $n_b$ is the number of droplets larger than the maximum allowed $D_{max}$. A breakage process is chosen when two or more droplets are larger than $D_{max}$. $P_{agg} = 1 - P_{brk}$. A random number $r_l$ is selected from a uniform distribution between 1 and 0. A breakage event is selected if $P_{brk} \geq r_l$, otherwise an aggregation event is selected. | [90,91,97] |

**Table 3.** *Cont.*

| Equation | Equations | Denote | References |
|---|---|---|---|
| (8) | $\frac{dN_s}{dt} = -0.16\alpha_{os}(D_s + D_o)^3(\varepsilon/v)^{\frac{1}{2}} N_s N_o$ | $D_s$ is sediment diameter. $D_o$ is oil droplet diameter. $N_s$ is OMA number concentration. $\alpha_{os}$ is the collision efficiency; $\varepsilon$ is the dissipation rate. $v$ is the kinematic viscosity of water. | [98] |
| (9) | $E = \frac{E_{max}\left(\frac{C_s}{C_{s50}}\right)^n}{1+\left(\frac{C_s}{C_{s50}}\right)^n}$ | $E_{max}$ is the maximum possible trapping efficiency. $C_s$ is the mass concentration of the sediment, $C_{s50}$ is the sediment concentration at 50% trapping efficiency. $n$ is the shape of the trapping efficiency versus sediment concentration curve. Least-squares fitting of the equation to the experiment data yields $n = 3$, and $E_{max} = 85\%$. | [99] |
| (10) | $E = \frac{(K_d/10^3)\times SPM}{1+(K_d/10^3)\times SPM}$ | $E$ is the oil trapping efficiency. $K_d$ is a distribution coefficient. $SPM$ is the sediment concentration. | [100] |
| (11) | $SR = \frac{V_s}{V} = \frac{r_c^{4-s}-r_{min}^{4-s}}{r_{max}^{4-s}-r_{min}^{4-s}} \times 100\%$ | $SR$ is the oil sin king ratio. $r_c$ is 50% of the critical oil droplets size. $r_{max}$ and $r_{min}$ are the maximum and minimum radii of the oil droplets. $s = 2.3$ based on laboratory data. | [101] |

Just like Sterling et al.'s model, breakup is neglected in several other OMA formation models [89,93,102,103]. For instance, Bandara et al. [89] developed a numerical model by using a three-dimensional (3D) advection-diffusion equation to simulate oil-sediment interaction (Equation (5) in Table 3). They used the Lagrangian Parcel (LP) method to reduce their program size and to operate more efficiently. Bandara et al.'s simulations showed that up to 65% of released oil can be removed from the water column as OMA. When oil droplets and sediment particles are smaller than 0.1 mm, more OMA form. Bandara et al. also stated that their lacking knowledge of oil sediment aggregation collision efficiencies, sediment aggregation efficiencies, and oil partitioning led to some uncertainty in their results, and more laboratory and field experimental work/data would further test and improve their model's adequacy [89]. Zhao et al. [93] developed the A-DROP model, based on the population balance equation, by introducing a new formulation of oil-mineral coagulation efficiency (Equation (6) in Table 3) to account for the coasted areas on the surface of oil droplets, the effects of hydrophobicity, and the ratio of particle to droplet size. The Zhao et al. formulation satisfactorily estimates oil trapping efficiency by incorporating the effects of shape and packing on OMA coagulation. They suggested that increasing particle concentration in the swash zone quickens oil–particle interaction, but the amount of oil trapped in the OMA does not correspond to the increasing particle concentration [93]. Recently, Cui et al. [102] developed an OMA formation model, OPAMOD, within the Coupled Ocean-Atmosphere-Wave-and-Sediment Transport (COAWST) modelling system, by modifying the existing population balance equation (Figure 3). The authors performed sensitivity tests on fractal dimension and collision efficiency by using the OPAMOD model. They stated that fractal dimension is important to OMA size distribution because fractal dimension influences the effective particle density; however, collision efficiency has less impact on the size distribution [102].

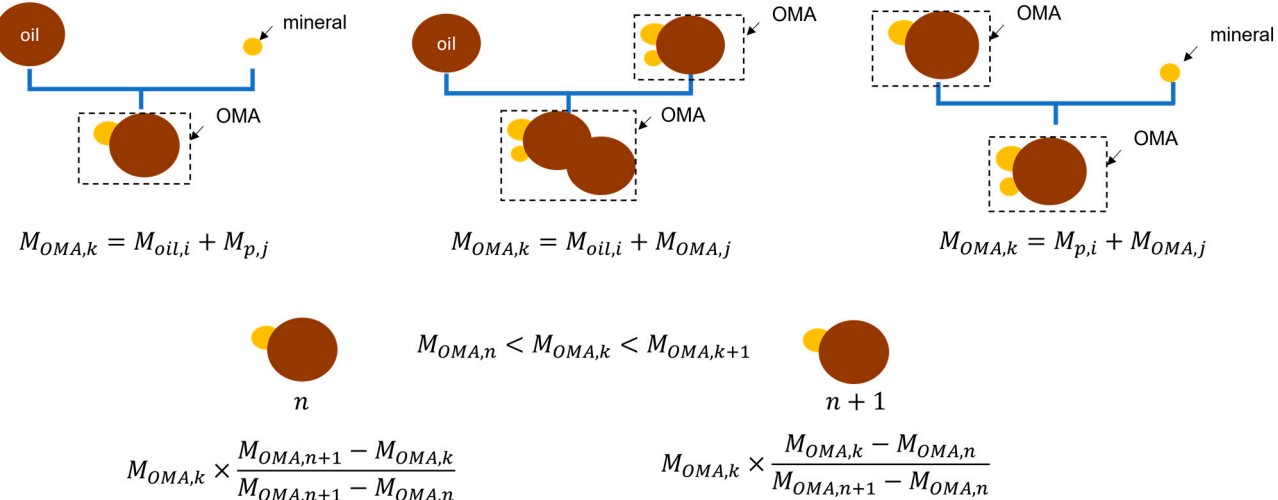

**Figure 3.** Schematic of OMA (OPA) formation model reproduced from Cui et al. (2021) [102], with kind permission Copyright from Frontiers Media S.A.

Although the size distribution of OMA is successfully simulated by the models that do not include disaggregation, the OMA breakup process is still valuable when modelling OMA formation. Khelifa et al. [90,91,97] developed a Monte Carlo simulation model (involving disaggregation) to predict OMA size distribution based on a population balance equation. The Monte Carlo method (Equation (7) in Table 3) is applied as a probabilistic tool to solve the model. The specific event, aggregation or disaggregation, is selected randomly at each step during the simulation. The simulation stops once equilibrium is reached. Khelifa et al. suggested that it is appropriate to integrate their new breakage model to describe oil droplet formation. The maximum permissible size of the oil droplets in their simulation is the key physical input of the model and can be upgraded by inte-

grating an empirical-theoretical model [97]. We note that this model predicts the particle size distribution well, but information about oil (mass) in OMA, needed for further risk assessment, is unavailable. In addition, the conceptual time to run the Khelifa et al. model using the Monte Carlo method is unknown.

In addition to the size distribution of the OMA, other parameters important to OMA formation, such as the time scale of formation [98,104], oil trapping efficiency [74,99,100], and oil sinking rate [101], have been studied and predicted. Hill et al. [98] presented Equation (8) (Table 3) to define the time required to form stable OMA [98]. They stated that OMA formation can occur over a short time scale. Among all their simulations, about 7% of the required times were shorter than 30 s, and 10% were longer than a day. In their modelling, when sediment concentration was high and droplets were large, the OMA formation time was the shortest. Hill et al. also suggested that their equation is suitable for the coastal zone [98]. Ajijolaiya et al. [99] developed Equation (9) (Table 3) to describe oil trapping efficiency. They found that trapping efficiency increases with improving sediment concentration and decreasing sediment size. Ajijolaiya et al. also stated that maximum trapping efficiency occurs when the range of sediment concentrations is near unity. Wang et al. [100] conducted several experiments and developed an empirical equation to calculate oil trapping efficiency as a function of sediment concentrations (Equation (10) in Table 3). Their simulation results indicated that the formation rate of OMA depends on sediment concentration, mixing time, salinity, and the use of a chemical dispersant in the water column. Wu et al. [101] developed a simple approach to estimate the sinking rate of spilled diluted bitumen (Equation (11) in Table 3). They suggested that the sinking rate is impacted by the density of oil, sediment, water, and oil size distribution.

*3.4. OMA Transport Modelling*

The transport of OMA is critically important to oil spill modelling. Some models have already been developed to simulate OMA transport [85,89,92,96,103,105–108]. The fate and transport of OMA have been typically modeled by implementing the advection-diffusion equation and a random walk model. For example, as we noted before, Bandara et al. [89] used the Lagrangian Parcel (LP) method to develop a model based on the 3D advection-diffusion equation to predict the fate and transport of OMA. The model simulates six processes: advection and diffusion of oil and sediments, dissolution of oil, aggregation of sediments, aggregation of oil-sediment particles, oil partitioning, and deposition of sediment and OMA. Unfortunately, Bandara et al. made some assumptions due to their lacking knowledge of oil sediment aggregation efficiencies, sediment aggregation efficiencies, and oil partitioning.

Niu and Lee [106] used a 3D random walk model to simulate the transport of OMA under hydrodynamic conditions involving wave-induced velocities, random velocities due to turbulence, and a settling velocity due to gravity (Figure 4). In their model, they use a fixed number of particles to represent the OMA at the spill site at the beginning of a simulation. Then, the particles move on each subsequent time-step, according to Lagrangian motion, while the OMA size distribution is artificially defined. Jones and Garcia [103] avoided having to artificially define the size distribution by combining the random walk model with the A-DROP model to simulate the fate and transport of OMA. The Jones and Garcia's model simulates the interactions between oil droplets and sediments during their residence time in a river, which informs the user under what conditions the oil is more likely to coagulate with suspended sediment. However, Jones and Garcia neglected the re-entrainment of settled OMA.

The settling velocity of OMA is one of the most important parameters when modelling OMA transport. There are equations that calculate settling velocity [89,93,96,106,109]. Winterwerp [109] developed the most widely used settling velocity equation basing it on Stokes' formula (Equation (12) in Table 4). Winterwerp's equation is suitable for spherical, massive particles in Stokes's regime ($R_e \leq 1$). This settling velocity equation integrates a particle tracking model or an advection-diffusion equation. Integrated model's simulation

results compare favorably with laboratory observations [89,110]. Later, Zhao et al. [102] improved the Winterwerp equation by expanding the applicable flows ($1 < R_e < 10^4$ and $R_e > 10^4$) as shown in Equation (13) (Table 4). Zhao et al.'s predicted settling velocity is also consistent with experimental data [103,111]. Niu et al. [103] developed a simple regression equation (Equation (14) in Table 4) to simulate the settling velocity relative to OMA diameter. Niu et al. also provide equations to estimate the concentration of the settled OMA and the oil content in each particle as shown in Equation (15) in Table 4 [106,112]. However, the Niu et al. equations are only suitable for specific conditions.

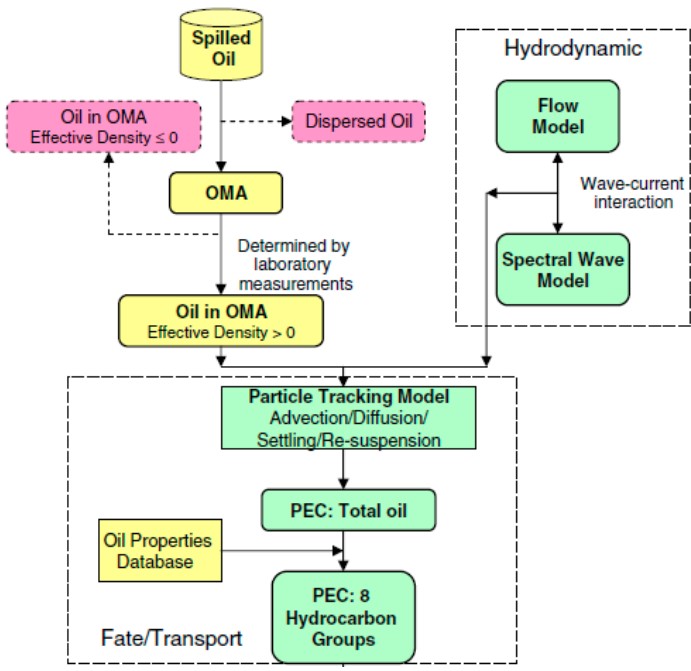

**Figure 4.** Flowchart of the modelling system adapted from Niu et al. [105] with kind permission from the authors.

**Table 4.** Equations used to estimate the settling velocity of oil-mineral-aggregate.

| Equation | Equations | Denote | References |
|:---:|:---:|:---|:---:|
| (12) | $w_s = \frac{(\rho_s - \rho_w)gD^2}{18\mu}$ | $w_s$ is the settling velocity of OMA. $D$ is the diameter of OMA. $\rho_w$ and $\mu$ is the density and viscosity of the liquid respectively. $\rho_s$ is the OMA density. and $g$ is the gravity. | [109] |
| (13) | $w_s = \sqrt{\frac{4g\|\rho_s - \rho_w\|D}{3C_D\rho_w}}$<br>If $R_e < 1$, $C_D = \frac{24}{R_e}$<br>If $1 < R_e < 10^4$, $C_D = \frac{24}{R_e} + \frac{3}{\sqrt{R_e}} + 0.34$<br>If $R_e > 10^4$, $C_D = 0.34\ to\ 0.4$ | $C_D$ is the drag coefficient. $R_e$ is the Reynold number. | [93,111] |
| (14) | $w_s = eD^f$ | $e$ and $f$ are regression coefficients. | [112] |

**Table 4.** *Cont.*

| Equation | Equations | Denote | References |
|---|---|---|---|
| (15) | $$C_{settled} = \sum_{i=1}^{k} \frac{N_i \times PM_i}{A_{cell}}$$ $$PM_i = \frac{M_{spilled} \times P_{settled} \times p_i}{n_i}$$ $$V_{oil} = \frac{\rho_{sediment} - \rho_{OMA}}{\rho_{sediment} - \rho_{oil}} V_{OMA}$$ | $k$ is the number of particle size classes. $N_i$ is the number of the $i_{th}$-size class particles in the user specified cell. $A_{cell}$ is the area of the cell, and $PM_i$ is the amount of oil per particle for the $i_{th}$-class particles. $M_{spilled}$ is the total mass of spilled oil. $P_{ssttled}$ is the percentage of spilled oil that may be transferred to sediment. $n_i$ is the number of particles used in the simulation for class $i$, and $p_i$ is the fraction of settled oil (in percentage) carried by the particle class. $V_{oil}$ is the volume of oil in an OMA of class $i$. $V_{OMA}$ is the volume of an OMA. $\rho_{oil}$, $\rho_{sediment}$, and $\rho_{OMA}$ are the densities of oil, sediment, and the OMA, respectively. | [105] |

## 4. Challenges and Recommendations

We described different factors that influence OMA formation and settling velocity, and the varied laboratory operations that different research groups conducted. Unfortunately, no one has conducted replicated experiments, and, thus, the experimental errors are unknown, making the reported data less comparable and reliable. Therefore, in the future, the experimental runs need to be replicated. Although many factors have been observed that affect OMA formation and settling, the significance of these factors remains unknown. We highly recommended designing a comprehensive statistical analysis that systematically investigates the significance of each factor on OMA formation and settling. In addition, effects of factors interacting with each other, such as dependencies, could exist. For example, the influence of temperature on OMA formation and settling could depend on the salinity. Identifying such dependencies is important for more accurate modelling of OMA formation and settling.

As for OMA formation modelling itself, oil-oil, sediment-sediment, OMA-OMA, oil-OMA, and sediment-OMA interactions are commonly neglected in current modelling. However, some of these processes could be important in predicting the OMA size distribution. In future modelling, we suggest including these interactive processes. Collision theory is widely applied to simulate OMA formation, and the collision efficiency/frequency is crucial to adequately simulating the process. However, available collision frequency/efficiency data are currently limited. More comprehensive empirical datasets are required. Moreover, although oil trapping efficiency is simulated by some of current models, the simulation results are barely validated due to insufficient experimental data. Current laboratory/modelling studies only focus on OMA or marine oil snow, but both OMA and marine oil snow likely cause oil sedimentation. Simultaneously studying OMA and marine oil snow deserves more research attention.

We also note that most OMA formation and transport models assume the oil droplets, sediment particles, and OMA are spherical. However, these particles' shapes vary significantly, and the shapes of the particles might be key to how OMA form and settle. Attempting to develop a shape factor for different particles is, therefore, meaningful. Lastly, an empirical equation for estimating an OMA's settling velocity is not always available, hindering the accuracy of OMA transport modelling. It is therefore valuable to combine experimental data with statistical regression to develop an adequate OMA settling velocity empirical equation.

## 5. Conclusions

Accurately tracking the transport and behavior of spilled oil is important to local authorities and oil companies because the knowledge makes the oil spill response more efficient and timelier, which reduces negative impacts. Numerous efforts have been made in oil spill modelling, in which oil-mineral-aggregate (OMA) formation, settling, and transport are modelled as crucial processes. Our review summarized oil spill weathering processes,

movement, and commonly used oil spill models. We further focused on current modelling of OMA processes. Briefly, we find:

- Many environmental factors have been reported to influence OMA size distribution and settling velocity, such as temperature, salinity, sediment concentration, the presence of dispersants, and so on.
- Statistical design and analysis should be used to determine the significance of each factor and their inter-dependencies.
- Attempts have been made to measure settling velocities in laboratory experiments, and the reported settling velocities ranged from 1 to 10.4 mm/s depending on experimental conditions. However, the lack of an adequate empirical equation for estimating an OMA settling velocity hinders OMA transport and oil spill modelling.
- Efforts at modelling the OMA size distribution have been made based upon collision theory. The Monte Carlo method has also been applied to model the size distribution of OMA; however, including the OMA breakup process or disregarding it could influence modelling results.

**Author Contributions:** Conceptualization, X.Z., Y.W. and H.N.; resources, L.L.; writing—original draft preparation, X.Z.; writing—review and editing, P.L. and H.N.; supervision, H.N.; project administration, H.N.; funding acquisition, H.N. All authors have read and agreed to the published version of the manuscript.

**Funding:** This research was funded by the Marine Environment Observation Prediction and Response Work (MEOPAR) [Project: 2-02-03-037.3] and Multi-Partner Research Initiative (MPRI) [Project: 6.02].

**Data Availability Statement:** Not applicable.

**Acknowledgments:** The authors acknowledge the Marine Environment Observation Prediction and Response Work (MEOPAR) and Multi-Partner Research Initiative (MPRI).

**Conflicts of Interest:** The authors declare no conflict of interest.

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
