# Peer review of "An Overview of Oil-Mineral-Aggregate Formation, Settling, and Transport Processes in Marine Oil Spill Models"

_jmse, doi:10.3390/jmse10050610_

Round 1
Reviewer 1 Report
The goal of a review article must be to comprehensively and completely reflect the current state of the scientific field.
Unfortunately, this main task was not fulfilled by the authors. In parts, the manuscript is sufficiently successful. For example, Chapters 3.1, 3.3, Table 3 and 4.
However, it is crucial that the major scientific advances of the last 10 years have not been sufficiently taken into account. The Deepwater Horizon accident has led to a large number of findings and publications. These are completely missing from this manuscript.
See, e.g.
Ainsworth eta, 2021 Ten Years of modeling the Deepwater Horizon oil spill, Environmental. Modeling & Software
French.Mc-Cay et al, 2021 Validation of oil Trajectory and fate Modeling of the Deepwater Horizon Oil Spill, Front. Mar. Science.
Particularly on the role of organic/biological material and dispengentia on transport processes, progress has been made to be noted.
As it stands, the manuscript is very incomplete and does not reflect the state of the art.
Author Response
We are grateful to the reviewer for his/her careful review. The comments and suggestions have substantially contributed to further improving the quality of the manuscript. According to the comments, great efforts have been made to revise the manuscript. The details regarding the response to the reviewers’ comments can be found in the attachment.

Reviewer 2 Report
The review article provides a comprehensive overview of research on the modelling of OMA. It will be appreciated by those who want to get a basic introduction and understanding of OMA, how OMA is used in oil spill modelling, and the current scientific literature on the topic (as given in the extensive reference list). This is a valuable addition to the scientific literature.
The article is organised well and is easy to follow. And despite the density of the topic, the article is not too long - an advantage for those who are over-extended in their work. The article would be improved with an additional English language edit, especially the Abstract and introduction.
Here are my few detailed comments.
- Please reconsider the title. The main focus of the article is on OMA modeling and other than Section 2.3 (commonly used models), oil spill modeling is not the focus. So the title starting with An overview of oil spill modeling, gives too much emphasis on the oil spill modeling. Consider changing the order, for example: An overview of oil-Mineral-Aggregate Formation, Settling, and Transport processes in oil spill models.
- End of Section 2.1 (lines 158-159): Is this referring to most 2D models or all models? If all models, then consider identifying which models in Table 1 do/do not include sedimentation processes.
- Section 3.1 (lines 184-188): This section on salinity is awkward. The focus of the paper is on oil spills in marine and coastal areas (see introduction) where salinity is at least 25 in the coastal zone away from river mouths and 35/36 in open waters. While the text discusses salinity effects at very low salinity. This is the exceptional case to the main information presented in the article. So the authors either need to write a sentence that prepares the reader for this or simplify this paragraph to say that within the normal range of salinities in open ocean/coastal areas, (lab) studies show no significant salinity effect (then add all the references).
- The inclusion of Table 3 is excellent and provides a great starting point for those wishing to develop new or improve existing models of OMA.
Author Response

(The authors gave the same response as above.)

Reviewer 3 Report
Please see the attached file.

Author Response

(The authors gave the same response as above.)

Reviewer 4 Report
The authors provided a review of various aspects of oil spills. They talked initially about oil spill models and listed them in a Table. Then, they addressed OMA formation, and they listed important equations in another Table. It would have been good if they took a position on papers instead of only reporting findings. For example, it is well established that the formation of OMA is enhanced by an increase in salinity from freshwater to seawater. The authors reported that but also that another researcher found
the opposite. This leaves the reader unsure about the next steps. The authors also addressed the equations that deal with the transport of OMA, and reported works such as Niu and Lee. The authors then got into Lyapnov exponents discussions and a related numerical method. I found this part to be alien to the rest of the manuscript. It is correct, but I am not sure it should be in this paper, but I leave this up to the Editor. The English needs improvement. It is not too bad, but it can benefit from some tweaks. Some comments are below.
“The most widely used settling velocity equation was developed by 402 Winterwerp (1998) based on Stokes’ formula (Eq. (1) in Table 4) [104]. This equation is 403 suitable for spherical, massive particles in Stokes’s regime
(??≤1). After that, this equa-404 tion was improved by integrating particle tracking model”
The authors are correct that the Stokes formula applies only to laminar flow, but they reported the generalized Stokes equation where the coefficient of drag Cd varies with the Reynolds number. However, they skipped from
laminar to fully turbulent without providing the Cd value for the intermediate range. They can show the formula for the full range from Zhao et al (2015, MPB). (Zhao et al. 2015)
“In the other words Should be “In other words”
It seems that the Lyapnov coefficients approach allows one to determine Lagrangian Coherent Cells, but this feature is only useful if it were under steady state conditions. If the ridges of high values vanish and move to
another location then the presence of a coherent cell at a given time is not of much use.
“On the other hand, the rest residuals are subject to naturally 112 disperse into the water column [27]. The oil emulsification depends” Maybe
“The remaining oil is subject to natural dispersion into the water column [27]. The oil emulsification depends” “movement process, which is mainly resulted from” Should be “movement process, which mainly results from”
“The spilled oil’s movement was primarily induced by surface currents when wind speed 133 < 10 km/h and the spill occur in locations close to land. Wind becomes a dominant factor 134 for the oil movement when its speed > 20 km/h and the spill happen in the open sea [23].”
Rewrite, wind is always important regardless of its speed. No need to make it either/or.
“However, Khelifa et al. (2008) observed that chemical dispersants can enhance the stickiness of oil and lead to larger OMAs [82].”
I have not read that paper, but the authors need to provide a justification for this controversial statement. Why would the dispersant enhance the stickiness of particles on oil?
Zhao, L., M. C. Boufadel, E. E. Adams, S. Socolofsky, T. king and K. Lee (2015). "Simulation of scenarios of oil droplet formation from the Deepwater Horizon blowout." Mar. Pollut. Bull. 101(1): 304-319.

Round 2
Reviewer 1 Report
authors corrected all comments.
The revised version is accepted
Author Response
Dear reviewer,
Thank you for your kind support and help with our work.
Best Wishes,
Xiaomei Zhong
On behalf of co-authors